# A Review of the Challenges in Deep Learning for Skeletal and Smooth Muscle Ultrasound Images

**Peter Ardhianto** [1,2], **Jen-Yung Tsai** [2], **Chih-Yang Lin** [3], **Ben-Yi Liau** [4], **Yih-Kuen Jan** [5], **Veit Babak Hamun Akbari** [6] **and Chi-Wen Lung** [5,6,*]

1   Department of Visual Communication Design, Soegijapranata Catholic University, Semarang 50234, Indonesia; peter.ardhianto@unika.ac.id
2   Department of Digital Media Design, Asia University, Taichung 41354, Taiwan; stevet@asia.edu.tw
3   Department of Electrical Engineering, Yuan Ze University, Chung-Li 32003, Taiwan; andrewlin@saturn.yzu.edu.tw
4   Department of Biomedical Engineering, Hungkuang University, Taichung 433304, Taiwan; byliau@hk.edu.tw
5   Rehabilitation Engineering Lab, Department of Kinesiology and Community Health, University of Illinois at Urbana-Champaign, Champaign, IL 61820, USA; yjan@illinois.edu
6   Department of Creative Product Design, Asia University, Taichung 41354, Taiwan; 109711569@gm.asia.edu.tw
*   Correspondence: cwlung@asia.edu.tw

**Featured Application: Deep learning is an effective strategy for determining skeletal and smooth muscle conditions to help clinic personnel in landmark identification, muscle site, and reliability testing using segmentation or classification via ultrasound images.**

**Abstract:** Deep learning has aided in the improvement of diagnosis identification, evaluation, and the interpretation of muscle ultrasound images, which may benefit clinical personnel. Muscle ultrasound images presents challenges such as low image quality due to noise, insufficient data, and different characteristics between skeletal and smooth muscles that can affect the effectiveness of deep learning results. From 2018 to 2020, deep learning has the improved solutions used to overcome these challenges; however, deep learning solutions for ultrasound images have not been compared to the conditions and strategies used to comprehend the current state of knowledge for handling skeletal and smooth muscle ultrasound images. This study aims to look at the challenges and trends of deep learning performance, especially in regard to overcoming muscle ultrasound image problems such as low image quality, muscle movement in skeletal muscles, and muscle thickness in smooth muscles. Skeletal muscle segmentation presents difficulties due to the regular movement of muscles and resulting noise, recording data through skipped connections, and modified layers required for upsampling. In skeletal muscle classification, the problems faced are area-specific, thus making a cropping strategy useful. Furthermore, there is no need to add additional layer modifications for smooth muscle segmentation as muscle thickness is the main problem in such cases.

**Keywords:** segmentation framework; classification method; network architecture; muscle disease; ultrasonography

## 1. Introduction

Muscle diseases are a significant public health issue and are known to increase mortality and morbidity risk factors [1]. Because of this risk, high rates of muscle disease investigation are common in clinical practice. Bruyere et al. described muscle problems in clinics, where 53.30% assessed muscle mass daily in clinical practice, 54.50% assessed muscle strength problems, and around 60% of geriatricians, endocrinologists, and rheumatologists estimated muscle mass [2]. The massive prevalence of muscle disease investigation in clinical practice can be supported with ultrasound devices, which offer responsiveness

and precision for detecting muscles disease in the range of 90 to 95% [3]. Muscle ultrasound images are a beneficial tool for investigating muscle disease in clinical and research environments [4].

Ultrasonography has transformative potential for measuring muscle health with the emerging interest in documenting and understanding muscle atrophy and function in every patient condition [5]. Additionally, ultrasound characteristics pose specific challenges in 1D [6], 2D [7], 3D [8], and 4D [9] images, such as image quality due to noise, scarce data, low image quality, and hand-held or operator skill, which may have a substantial influence on the efficacy of deep learning performance when compared with X-ray, computed tomography (CT), or magnetic resonance imaging (MRI) images [10] due to the special attention and resources required in the deep learning process. Furthermore, the challenges in the muscle ultrasound imaging process include the identification of landmarks, reliability testing, muscle site tracking, image acquisition and analysis, equipment use, normative data, and the interpretation of results [11,12]. The limitations of muscle ultrasound imaging can be overcome with good training for clinical personnel and clinical experience [4], and such practice is common in clinics and is painless and noninvasive [13]; however, it is costly and time consuming. Deep learning is an upcoming solution for the detection of landmarks [14], muscle site tracking, reliability checking, and the analysis and interpretation of ultrasound images which may support novice clinical personnel [15,16]. Deep learning has an advantage over other classification and segmentation strategies featuring artificial intelligence in that the efficiency of segmentation can be greatly improved through different sets of filters that can be implemented for each stage in a convolutional neural network (CNN) based and an overall deeper network structure. Methods supervised by deep learning methods are expected to be superior because they use pixel-level annotation, which provides crucial information for dealing with various intensity patterns, especially at the edges of muscle ultrasound images [17–19].

Deep learning in ultrasound images has been used to support clinical tasks since 2015 [20], such as identification of the abdominal plane [21,22], frame labeling on fetal pregnancy images [23], analyzing abdominal circumference [24], vessel detection [25], tongue contour extraction [26], carotid plaque classification [27], thyroid nodule detection [28], and the identification liver cancer [29]. Studies on deep learning ultrasound images with organs, the thyroid, breast, liver, heart, bone, and brain can be found in the literature [20]; however, few studies in the literature on deep learning muscle ultrasound images in skeletal and smooth muscles describe or summarize the strategies required to overcome the inherent challenges.

In 2018 (Figure 1), deep learning was increasingly used to support skeletal muscle and smooth muscle ultrasound imaging in order to improve reliability testing for the classification of muscle types [30], classifying muscles by gender [31], and the classification of muscle vibration [32]. Additionally, it supported the identification of landmarks such as segmentation in the orientation of muscle fibers [33] and tracking the cross-sectional area of the rectus femoris [34]. In addition, deep learning methods have visualized the neck muscle pattern landmark and muscle–tendon landmark [35,36]. Besides, deep learning has assisted measurement and tracking in the urogenital hiatus and puborectalis muscle [37]. Moreover, deep learning methods have been evaluated regarding the ciliary muscle for glaucoma interpretation and measured bladder wall thickness [38,39]. Deep learning will be especially beneficial when considering ultrasound muscle images with different types of muscles, namely skeletal muscles, smooth muscles, and cardiac muscles [40,41].

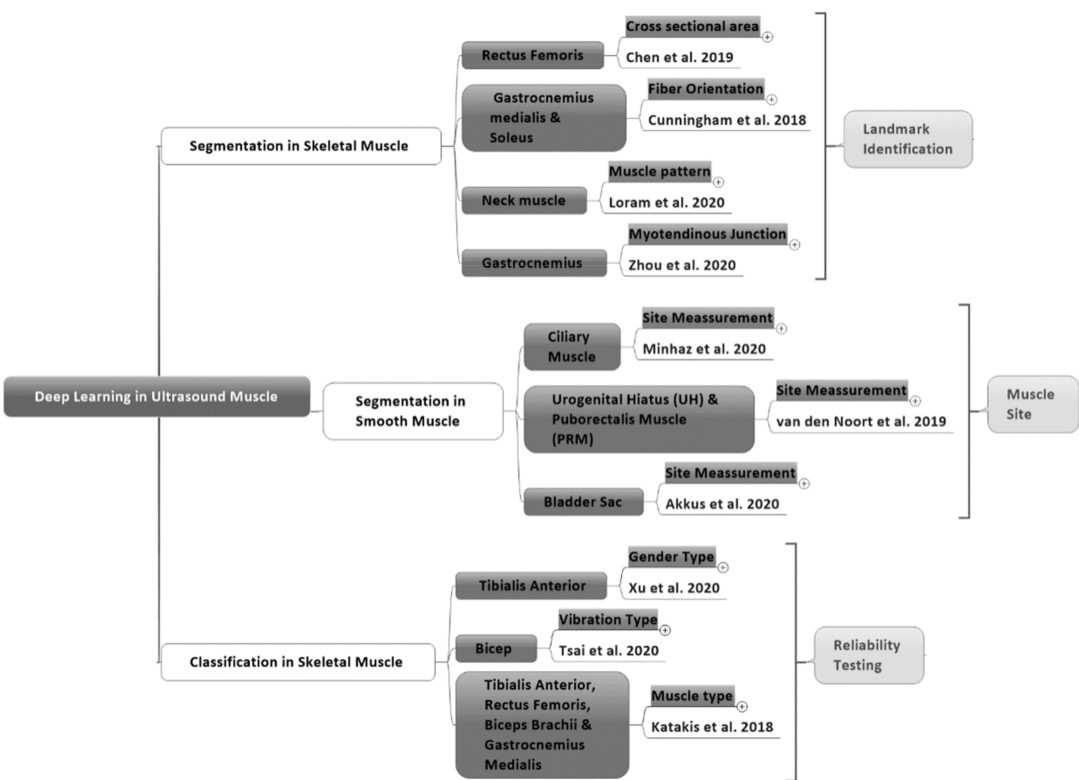

**Figure 1.** The purposes of deep learning in studies of skeletal and smooth muscle areas from 2018–2020.

As far as we know, there has been no discussion about cardiac muscle tissue in deep learning from 2018 to 2020. Cardiac muscles are one of three types of muscle, with the other two being skeletal and smooth muscles [42]. We found seven studies on skeletal muscles and three studies on smooth muscles from 2018 to 2020 which had case conditions and deep learning performance parameters. For skeletal muscle ultrasound imaging, image noise and muscle movement require special treatment for deep learning architecture and segmentation [30,41]. Smooth muscle research has focused on measuring muscle thickness [38]; however, the treatment of image problems in skeletal muscles and smooth muscles has not been compared in order to understand the current state of knowledge. Therefore, it is necessary to comprehend the comparison of deep learning conditions and strategies between ultrasound muscle images.

This study aims to look at the trends and challenges of deep learning to overcome muscle ultrasound imaging problems from 2018 to 2020 by comparing the conditions and strategies in skeletal muscles and smooth muscles to assess the current state of knowledge. This study will enhance knowledge of the deep learning conditions and strategies for handling muscle ultrasound images.

## 2. Methods

### 2.1. Analysis of the Literature

We performed a thorough analysis of literature using the Scopus, Web of Science, and PubMed search engines between 2018–2020, which was conducted in July 2020 with the keywords "deep learning" and "ultrasound" and "muscle" and found 83 studies in total. We included 10 peer-reviewed journal publications and conference proceedings. The Journal of Ultrasound Medical, Journal of Imaging, Journal of the Royal Society Interface, Sensors Journal, Ultrasound Obstet Gynecol, Biomedical Signal Processing and Control, IEEE Journal of Biomedical and Health Informatics, SPIE Medical Imaging, International Conference on Applied Human Factors and Ergonomics, and IEEE Conference Proceedings describe the application of deep learning in muscle ultrasound imaging. All search results

were imported into EndNote Reference Manager Version 9.1 to collect and automatically detect duplicate data (Figure 2).

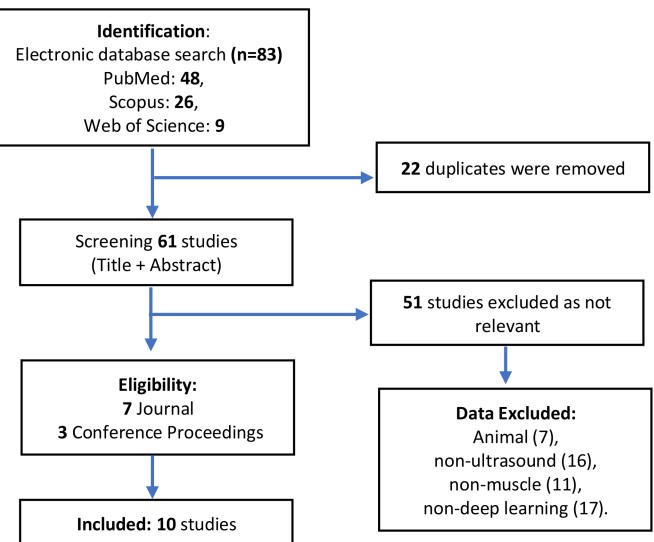

**Figure 2.** A review of deep learning in muscle ultrasound imaging.

### 2.2. Data Eligibility Requirement

Animal, non-muscle (such as bone, organ, tissue, thyroid, cancer, and tumor), non-deep learning (such as machine learning and big data), and non-ultrasound (such as MRI, CT-Scan and X-Ray) studies were excluded. Reviews, case studies, letters, opinions, and mini-articles were exempted. Two researchers separately analyzed the results and addressed discrepancies by discussion (P.A. and C.L.). The abstracts and full texts of the related papers were thoroughly read, and only those that met the requirements were chosen. There were 10 studies that met the criteria in the results. The researchers then confirmed the selected papers and addressed any disagreements; if any disagreement remained, a third researcher was invited to discuss it and the findings were evaluated (Y.L.). There was no disagreement in the selection of papers in this review.

### 2.3. Data Analysis

The findings were divided into three groups: skeletal muscle segmentation, smooth muscle segmentation, and skeletal muscle classification (Figure 1). Besides, the following data are summarized: (1) the muscle type, which provides information about the skeletal or smooth muscle type; (2) the deep learning approach for the segmentation or classification of ultrasound images; (3) citations, containing author information and the year the article was published; (4) the research objectives that cover specific muscle areas; (5) the deep learning architecture method; (6) the dataset as a reference for a deep learning model studying ultrasound images; (7) the output regarding matrix performance to view the deep learning results; and (8) image solutions to determine the method used to solve problems in muscle ultrasound images to improve deep learning performance from 2018 to 2020 (Table 1).

**Table 1.** Deep learning approaches with image solution strategies to handle skeletal and smooth muscle ultrasound image problems. (CNN: Convolutional Neural Network, DCNN: Deconvolutional Neural Network, RAN: Region Adaptive Network, Deep-CNN: Deep Convolutional Neural Network, MD: Mean Difference, MAE: Mean Absolute Error, RMSE: Root Mean Square Error, HD: Hausdorff Distance).

| Muscle Type | Deep Learning Method | Citation | Purpose | Method | Dataset (Images) | Output | Image Solution |
|---|---|---|---|---|---|---|---|
| Skeletal | Segmentation | Chen et al., 2019 [34] | Cross-sectional area Rectus femoris | CNN | 81,600 | Dice: 0.90 ± 0.02 Precision: 0.93 ± 0.02 Recall: 0.88 ± 0.04 | 180° rotation |
| | Segmentation | Cunningham et al., 2018 [33] | Fiber orientation in medial gastrocnemius and soleus | DCNN | 1008 | MD 3.27° ± 10. MAE 10° ± 8.09° RMSE 10.62° | −5° +5° rotation |
| | Segmentation | Zhou et al., 2020 [36] | Tracking the myotendinous junction in the gastrocnemius | RAN | 300 | Dice 80.1% | Transfer Learning |
| | Segmentation | Loram et al., 2020 [35] | Objective visualization and pattern analysis neck muscle | U-Net | 3272 | Dice 64 ± 21% HD 5.7 ± 4.0 mm Precision 94 ± 3% | −8° +8° rotation |
| | Classification | Xu et al., 2020 [31] | Gender classification in the tibialis anterior area | CNN (VGG-16) | 115,916 | Accuracy: VGG-16 95.2% (Alex Net 83.3% Google Net 84.4%) | 4 Random rotation angles and flipping |
| | Classification | Katakis et al., 2018 [30] | Muscle type of the rectus femoris, tibialis anterior, gastrocnemius medialis and bicep brachii | Deep-CNN (VGG-F) | 300 | Accuracy: VGG-F 89.4% (VGG-16 87.5%, VGG-19 87.2%) | Transfer learning |
| | Classification | Tsai et al., 2020 [32] | Bicep vibration muscle | CNN (AlexNet) | 120,423 | Accuracy: AlexNet 82.5% (VGG-16 79.9%, VGG-19 79.7%) | 45° rotation, flipping, adjust contrast |
| Smooth | Segmentation | Minhaz et al., 2020 [39] | 3D assessment ciliary muscle | U-Net | 1000 | Dice 0.72 ± 0.070 Accuracy 0.93 ± 0.010 Sensitivity 0.79 ± 0.070 | No |
| | Segmentation | van den Noort et al., 2019 [37] | Area measurement in urogenital hiatus and puborectalis muscle | CNN | 713 | Dice 0.94 ± 0.020 and 0.73 ± 0.090 | No |
| | Segmentation | Akkus et al., 2020 [38] | Fully automated segmentation Bladder wall thickness. | U-Net | 800 | Dice 0.93 ± 0.040 mm RMSE 0.70 ± 0.20 mm | 10 times random elastic deformation |

## 3. Results and Discussion

### 3.1. Numbers of Studies

The two deep learning approaches, classification and segmentation, were found in 10 studies on ultrasound muscle images. In 2018, there were two studies on segmentation and one study on ultrasound muscle classification. In 2019, there was only one study on segmentation. Moreover, in 2020, muscle images on ultrasound using deep learning increased to seven studies (five studies on segmentation and two studies on classification). As a result, in deep learning studies on ultrasound muscle images, studies on segmentation dominated with seven studies compared to studies on classification with three studies (Figure 3).

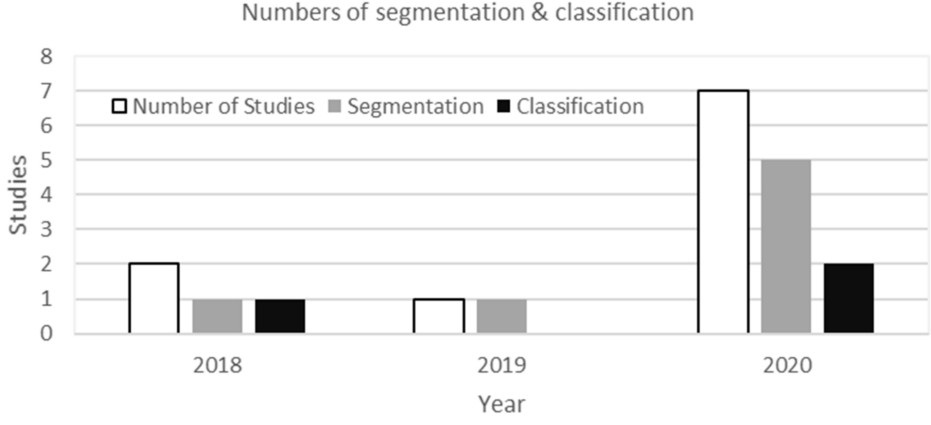

**Figure 3.** The development of deep learning studies for muscle ultrasound images from 2018 to 2020.

### 3.2. Network Architecture

Based on ten studies on deep learning on ultrasound muscle images, it was found that the U-Net and CNN architectures were the most used network architectures in segmentation [34,35,37–39]. Meanwhile, the RAN was the latest development of the region-based convolutional neural network (RCNN) [36]. The classification entirely uses a CNN based on AlexNet, Deep-CNN, and VGG [30–32] (Figure 4).

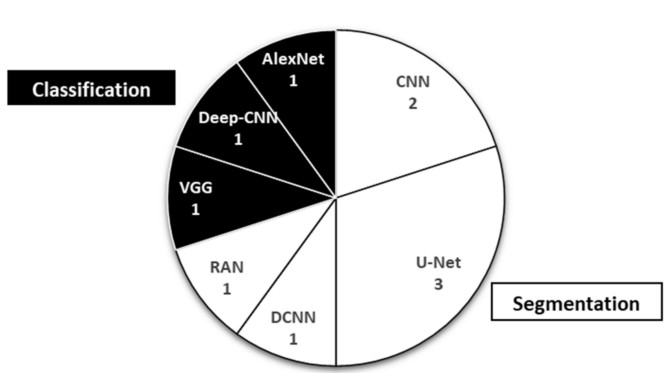

**Figure 4.** Various network architectures used in deep learning studies on muscle ultrasound images

### 3.3. Segmentation in Skeletal Muscle

Skeletal muscle ultrasound imaging is the largest area of deep learning research regarding muscle ultrasound images [20]. There were seven studies on the use of deep learning with two purposes, with four studies on segmentation and three classification studies. In the segmentation of skeletal muscle ultrasound images, the output was helping clinics with landmark identification. The problems in skeletal muscle tissue, in general,

are noise on the image and are related to movement [30]. Because of these problems, for segmentation in skeletal muscle images, it is necessary to modify the network architecture to achieve the maximum performance results. The Dice similarity coefficient is common to be used in segmentation task [19]. The Dice similarity coefficient is also considered superior because it does not only measure the proper marking of the number of pixels but also the precision of boundary segmentation [43].

### 3.4. Segmentation Architecture in Skeletal Muscle

The network architecture used for the skeletal muscle segmentation requires specific layer treatments to maximize deep learning performance. It is necessary to combine three different layer types in the network architecture for upsampling with a feature representation phase and score map reconstruction phase in segmentation (Figure 5A). Chen et al. (2019) built a CNN to segment the rectus muscle using a deconvolutional layer, a concatenation layer, and a convolutional layer plus a rectified linear unit (ReLU) activator. The convolution layer was the core building block of the CNN. The layer's parameters consisted of a small number of learnable filters (or kernels) that were wide and deep [44]. The CNN network consisted of two stages: the feature map extraction function and the reconstruction of the score map. The deconvolutional layer multiplies a single input with a learned filter to generate an output patch. In this situation, the patches that overlap in the output are simply added to create the result [45]; however, the deconvolution would cause the loss of the high-resolution information in images [34,46]. The concatenation layer connects the enlarged score map with the corresponding feature map. It fuses the feature maps in the previous pooling layers or convolution layers with the current feature maps in the deconvolution layer to overcome high-resolution information loss [47,48]. The skip connection layer design could capture more multiscale contextual information to improve segmentation accuracy [19]. The convolution layer integrated the image information after the concatenation layer. The performance of the proposed model was evaluated and compared to state-of-the-art muscle segmentation methods. During voluntary contraction, a real-time ultrasound image sequence was obtained from the rectus femoris muscle to establish an automated tracing method for the cross-sectional area and shows high matrix performance.

A deconvolutional layer without a concatenation layer can be used for segmentation in skeletal muscles [49] (Figure 5B,D); however, when using the deconvolutional layer in upsampling, it needs to go through the fully connected layer between the convolutional layer and the ReLu activator to consider cropping strategies for the preprocessing. Cunningham et al. used a DCNN in 2018 to look at the shape of a fiber pathway and muscle location with a fully connected layer to combat overfitting using DCNN architecture [33]. Like Cunningham et al., Zhou et al. determined the mechanics and pathological conditions of a muscle–tendon complex in 2020 and monitored the myotendinous junction in ultrasound images using a RAN. The RAN used a deconvolutional layer without a concatenation layer [36]; however, the RAN was not fully connected using the cropping strategy for regions of interest (ROI) in preprocessing. The ultrasound image on the myotendinous junction had characteristic distribution data with noise in different slice hyperplanes which can reduce segmentation accuracy. Therefore, a U-shaped segmentation network in each layer consists of the convolutional layer, followed by a batch normalization layer and a ReLu [50,51]. In skeletal muscle ultrasound imaging, the use of a RAN presents advantages in applications that use single-shot techniques and transfers learning to solve the inherent muscle ultrasound imaging problems. A RAN was proposed with a unified framework, transfer learning, and region cropping strategies to address the challenges of poor image quality, scarce data, and data imbalance; however, RANs are not currently used in areas other than CNNs, which are often used in the context of skeletal muscles.

A) Chen et.al. (2019). CNN Architecture for Rectus Femoris Segmentation

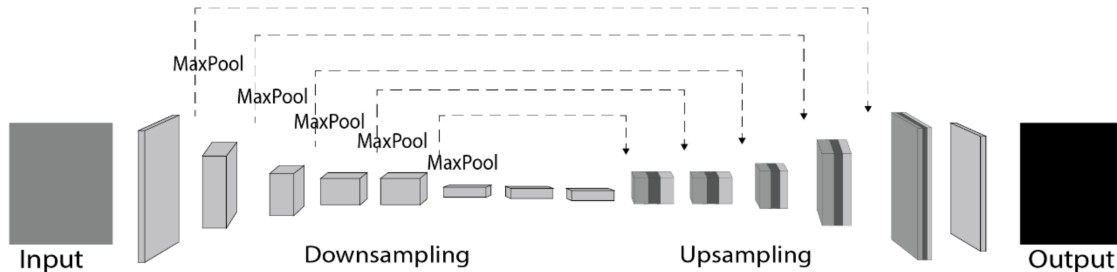

B) Chunningham et.al. (2018). DCNN Architecture for Fiber Orientation

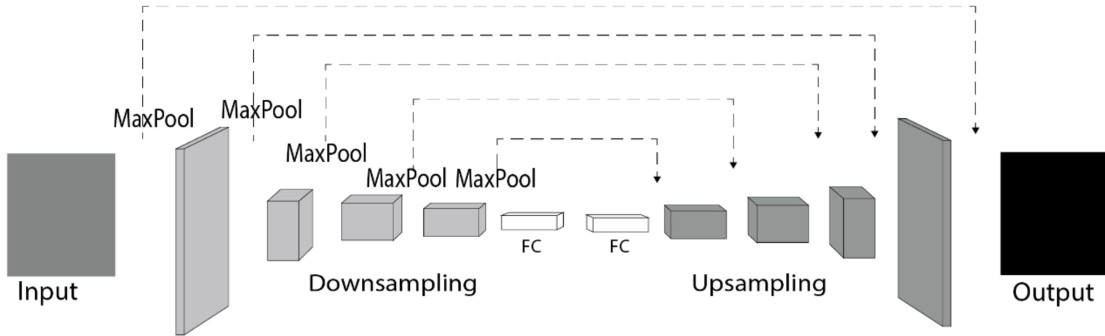

C) Loram et.al. (2020). U-Net Architecture for Neck Pattern

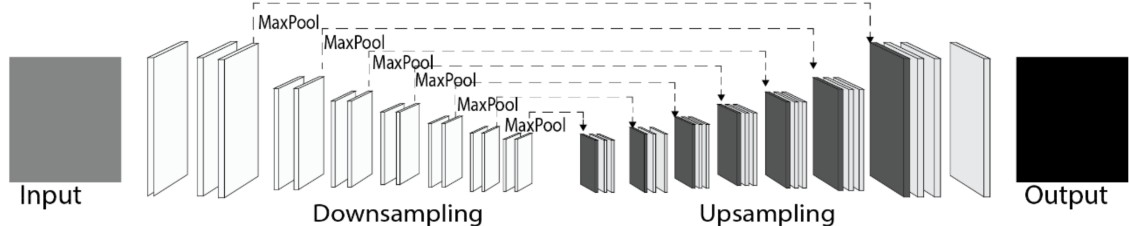

D) Zhou et.al. (2020). RAN Architecture for Myotendinous Junction

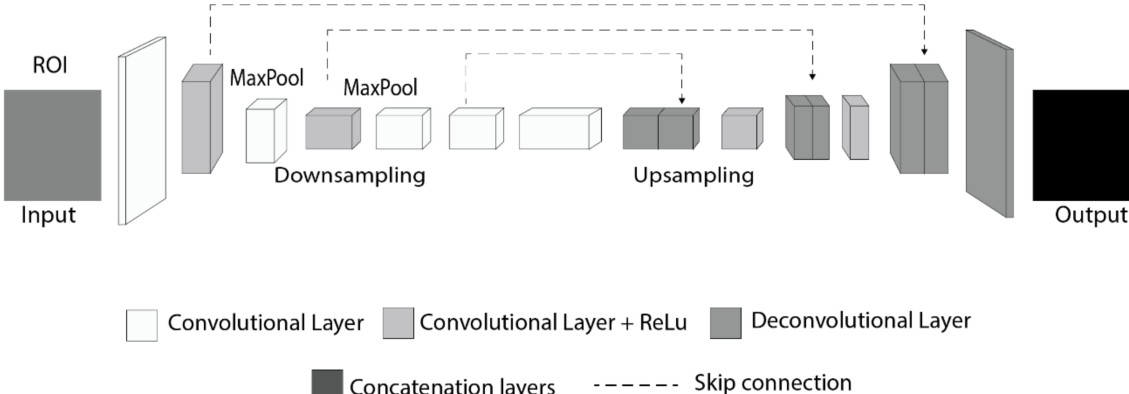

**Figure 5.** Segmentation in skeletal muscles; (**A**) CNN architecture with three different layers in the upsampling for rectus femoris segmentation. (**B**) DCNN architecture with fully connected layer for segmenting fiber orientation. (**C**) U-Net architecture with two different layers in upsampling for neck muscle pattern segmentation. (**D**) RAN architecture with two different layers for downsampling and upsampling for tracking the myotendinous junction.

In a different skeletal muscle area, the concatenation layer without the deconvolution layer is possible for segmentation in skeletal muscles to preserve image details using the concatenation layer and convolutional layer with ReLu (Figure 5C). Loram et al. (2020) used U-Net to find precisely identify a detailed location and see the longitudinal record effect on individual neck muscles. Deep learning with concatenation layers followed by a convolutional layer plus ReLu in upsampling U-Net network architecture was used to ensure image detail for the same injection site in [35]. In this case, the deep learning precision outcome is the main factor that represents image detail as clinical personnel can see the impact of health progress after continuous injection at an exact location. A concatenation layer is required to improve precision sharpness, which is followed by a convolution layer to acquire the final output to overcome the imaging problem [52]. This strategy can help increase the precision of feature identification and decrease the degree of information loss [35,53,54].

Two phases and skip connections are always used in the skeletal muscle segmentation (Figure 5). Two phases are used because the ultrasound image has a noise problem, while the area to be segmented is small and close to other muscles that move constantly. Additionally, the use of two phases is an effort to maintain the accuracy of segmentation for muscle identification supported by using a skip connection to eliminate singularities [55,56]. Skip connections are additional connections between nodes in different layers of the neural network [57]. Skip connections can operate with very few training images and yield precise segmentation [36]. Skip connections are essential for preserving boundary details when dealing with limited image quality [58]; however, the resulting network architecture has different scenarios as the network architectures are specific for their given purposes and dataset; however, data imbalances and scarce images are common occurrences with skeletal muscles. The popular solution to solve data imbalances and limited numbers of images is to use an augmentation method with image rotation [30,31,33–35,39] where the orientation ranges from 5° to 180°. Apart from rotation and image flipping, transferred learning is used to solve image quality problems and data imbalances in skeletal muscle segmentation applications.

### 3.5. Segmentation in Smooth Muscle

Meanwhile, in smooth muscles, the general challenge is to measure the thickness of a specific muscle type rather than a complete skeletal muscle [38]. Smooth muscles are located on internal organs and do not have transverse lines, unlike skeletal muscles [59]. The network architecture used in smooth muscle segmentation does not require modifications such as those with skeletal muscles.

### 3.6. Segmentation Architecture in Smooth Muscles

CNN and U-Net can be used for smooth muscle segmentation. Akkus et al. (2020) and Minhaz et al. (2020) used U-Net with a convolutional layer and ReLu with biomedical images. Both were created [60] to perform muscle thickness segmentation with a bladder wall thickness layout and to identify the best possible location [38]. This was performed alongside the use of segmented 3D ultrasound images of the ciliary muscles [39]. This study found high accuracy and was the only study with 3D images produced by muscle ultrasound imaging from 2018 to 2020. Furthermore, van den Noort et al. (2019) used a CNN to measure the area of the urogenital hiatus and puborectalis muscle by automated segmentation [37]. The CNN was used with a self-normalizing neural network. A self-normalizing neural network is used to reduce network complexity and use a smaller mini-batch size without sacrificing the generalization performance [61]. The study showed the highest Dice result for the urogenital hiatus matrix (Table 1).

Different from skeletal muscles, smooth muscles rarely have problems with imbalances and scarce data. Of the three cases, there is one study that used augmentation to solve image scarcity with smooth muscles. Akkus et al. had difficulties with the amount of image data, requiring data augmentation with a ten times random elastic deformation

approach. Adding the amount of images and increasing the measurements improved the performance of deep learning for the bladder wall muscle thickness [38]. Two studies used a sufficient dataset, where Minhaz et al. (2020) used 3D ciliary muscle images with 1000 images, and deep learning achieved high accuracy without a scarce data solution [39]. Additionally, van den Noort et al. produced measurements with 2D images with a CNN with the urogenital hiatus and puborectalis muscle [37] with 713 images and achieved the highest Dice coefficient without any scarce data solution.

### 3.7. Classification in Skeletal Muscle

Based on the reviews shown in Table 1 for the classification of muscles using deep learning from 2018 to 2020, three network architectures were useful for classifying skeletal muscle. The use of augmentation, rotation, and transferred learning is needed to solve problems in skeletal muscle area data. Besides, cropping strategies need to be considered to classify skeletal muscles, especially in the calf muscle area.

### 3.8. Classification Architecture in Skeletal Muscle

CNNs and deep CNNs have been commonly used as base architectures for the classification of muscle ultrasound images [20]. Specifically, VGG-16 and AlexNet are variations of a CNN which are used to classify muscle gender and bicep vibration, while VGG-F is a variation of a deep CNN which is used to classify calf muscle types. The difference between the two lies in the number of layers. The deep CNN uses five convolutional layers followed by two fully connected layers and is useful for classifying many labels. For example, Katakis et al. (2018) classified eight classes. Meanwhile, CNN-based AlexNet and VGG-16 were used to classify two and three classes, respectively.

VGG-16 was used for gender classification in the anterior tibialis area. The VGG-16 accuracy was 95.2% compared to 83.30% for AlexNet and 84.40% for GoogleNet in previous trial studies [31]; however, the training data weakness arises from the fact that the full-frame ultrasound images caused inaccuracy, namely 100% for males and 49.30% for females, thus requiring cropping strategies to perform more detailed classification and eliminate the lower part and right side. The same problem occurred in classification with VGG-F when classifying muscle types in the transverse and longitudinal planes in the musculoskeletal area [30]. A cropping strategy was implemented by excluding the fasciae and muscle tendons, where only the muscle mass in the fiber area was classified. VGG-F outperformed VGG-16 and VGG-19 for muscle type classification with an accuracy of 89.44%. A cropping strategy can help a classification process to gain more leverage for specific area identification [62,63].

In contrast, with deep learning classification of bicep vibration, the cropping strategy was not used because it requires an entire image to identify the difference in muscle thickness on the bicep vibration. AlexNet provided 82.50% of the results and is comparable with VGG-16 and VGG-19 for the test results with the same number of images [32].

### 3.9. Dataset Requirement

Referring to Table 1, the ranges used for the dataset in the training process with the ultrasound muscle images are 300 to 120,423 images. The minimum dataset without an image data solution was shown by van den Noorth et al. (2019) [37] with 713 images for measuring the area in urogenital hiatus and puborectalis muscle with CNN segmentation. The largest dataset without an image solution featured 1000 images for 3D image assessment with the U-Net architecture approach for segmentation [39]. The datasets used in the ten studies were derived from capturing B-mode ultrasound images. In addition, the use of segmentation in thyroid nodules can be used in real-time [64]. This allows the possible application of real-time deep learning with muscle ultrasound images.

Difficulties with the number of images were common challenges among the two studies which did not use augmentation or transferred learning techniques with the muscle ultrasound images. In comparison to other artificial intelligence techniques, deep learning

requires a higher amount of data, therefore the data quantity required for training needs to be known. Furthermore, data augmentation would not support geometries such as 3D images [65], except for automatically segmenting 3D volumes of biomedical images, where multidimensional gated recurrent units are used as the main layers of the neural network model. The method proposed involved on-the-fly data augmentation [66,67] or manual augmentation of data in robust 3D models with different orientations and sizes [68]; however, gated recurrent units and automatic on-the-fly data augmentation requires further investigation in the context of muscle ultrasound images.

### *3.10. Matrix Performance Result*

In the segmentation, the most popular matrix performance output was the Dice coefficient. The highest Dice value was found for UH segmentation with a gain of $0.94 \pm 0.02$ with a CNN network architecture with smooth muscles, while the highest precision was that of the cross-sectional area of the rectus femoris with a value of $0.93 \pm 0.02$ based on a CNN architecture with skeletal muscles. Furthermore, the accuracy results were the most popular for classification. VGG-16 shows the highest results for gender classification with 95.20% accuracy.

### *3.11. Limitation*

There are two limitations in this study. The first limitation is that the muscle segmentation and classification studies use data augmentation or transferred learning because of the inherent muscle ultrasound image problems; however, both solutions currently still have disadvantages and need further investigation [69]. Future work will discover additional solutions for muscle ultrasound image problems, such 4D ultrasound, on-the-fly data augmentation for 3D images, and active learning as an alternative for augmentation or transferred learning to overcome such limitations [70,71]. The second limitation is that this study is limited by two types of muscles: skeletal muscles and smooth muscles, without examining cardiac muscles. The difficulty with cardiac muscle ultrasound imaging is the sequential nature of muscle data. A deep learning model which can read sequential data is needed, such as a long short-term memory (LSTM) model [72,73].

## 4. Conclusions

Deep learning is an effective strategy for determining skeletal muscle and smooth muscle conditions to help clinical personnel in landmark identification, muscle site identification, and reliability testing using segmentation or classification via ultrasound imaging. The inherent challenge relating to skeletal muscle segmentation is that such muscles move regularly and that this produces noise, thus skipped connections and modified layers in upsampling are needed to ensure that data are recorded. There is no need to add additional layer modifications for smooth muscle segmentation because muscle thickness is the main problem. In skeletal muscle classification, the problems faced are area-specific, thus making a cropping strategy useful. The use of segmentation with B-mode ultrasound imaging allows the possible application of real-time deep learning to muscle ultrasound images.

**Author Contributions:** Conceptualization, P.A. and C.-W.L.; methodology, C.-Y.L. and J.-Y.T.; writing—original draft preparation, P.A.; writing—review and editing, C.-W.L. and V.B.H.A.; supervision, B.-Y.L. and Y.-K.J. All authors have read and agreed to the published version of the manuscript.

**Funding:** This study was supported by the Ministry of Science and Technology of the Republic of China (MOST 108-2221-E-241-008, MOST-108-2221-E-468-018), Kuang Tien General Hospital and Hungkuang University (HK-KTOH-109-04), and Asia University Hospital and China Medical University Hospital (ASIA-107-AUH-09).

**Institutional Review Board Statement:** Not applicable.

**Informed Consent Statement:** Not applicable.

**Data Availability Statement:** The data used to support the finding of this study are included within the article.

**Acknowledgments:** The Authors wish to express gratitude to Fityanul Akhyar, and Syauki Aulia Thamrin, for their assistance.

**Conflicts of Interest:** The authors declare that they have no known competing financial interests or personal relationships that could have appeared to influence the work reported in this paper.

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
