# Peer review of "A Review of the Challenges in Deep Learning for Skeletal and Smooth Muscle Ultrasound Images"

_applsci, doi:10.3390/app11094021_

Round 1

Reviewer 1 Report

See attached file 

Author Response

General Concerns

[1]. Timeline

This paper sets out to review the application of Deep Learning to the processing of ultrasound images of muscle. Whilst this aim is laudable the value of it must be questioned with only 10 studies identified. Given this, there is a need to properly justify the timeliness of the review and what is to be achieved by it.

## Authors’ Response:

Thank you for mentioning timelines as a reason for selecting the 10 studies. We agree and consider it is essential to strengthening our study. We have revised and more emphasized the timeline as shown below:

*** Page: 2, line 68-69 and *** Page: 3, line 82-83

Deep learning in ultrasound images has been used to support clinical tasks, starting in 2015 [20],”

“In 2018 (Figure 1), deep learning has been increasingly used to support skeletal muscle and smooth muscle ultrasound to improve reliability testing.”

[2]. Advantages

With only 10 studies in the review an obvious question that needs to be addressed is: what are the advantages of deep learning techniques over other classification and segmentation techniques, including AI techniques that would not be classified as deep learning?

## Authors’ Response: Thank you very much for your thoughtful review of our manuscript. We have added the references to explain the advantages of deep learning in medical images over the other AI’s classification and segmentation techniques in the introduction section as shown below.

*** Page: 2, line 61-66

“Deep learning has an advantage over other classification and segmentation strategies in artificial intelligence in that the efficiency of segmentation can be greatly improved through different sets of filters that can be implemented for each stage in a CNN-based network and deeper network structure. Deep learning supervised methods are expected to be superior because they use pixel-level annotation, which provides crucial information for dealing with various intensity patterns, especially at the edge for muscle ultrasound images [17-19].”

[3]. Inappropriate

Some of the language used is inappropriate for example describing a technique as ‘…widely used…’ (line 147) when it was used in only 3 of 10 studies.

## Authors’ Response: Thank you for the reviewer’s has reminded us about using the inappropriate describing. We agree with the reviewer's suggestion and have revised it.

[4]. Challenges

Whilst the section on dataset requirements (section 3.9) is helpful, the title suggests that the challenges in applying deep learning techniques to muscle ultrasound imaging will be a key part of the review; in reality this does not form part of the review but rather the review focusses on how specific deep learning techniques have been applied to selected problems in the area.

## Authors’ Response: Thank you for the reviewers' comments. We consider that challenges in applying deep learning techniques to muscle ultrasound imaging are important things that we have to explain. We revise section 3.9 as below:

*** Page: 10, line 335-344

“The dataset used of the ten studies was derived from capturing Ultrasound B-Mode images. In addition, the use of segmentation on ultrasound B-Mode in thyroid nodules can be used in real-time [66]. It allows the possible application of real-time deep learning to muscle ultrasound images.

Difficulties with the number of images were common challenges among the two studies which did not use the augmentation or transfer learning technique in muscle ultrasound images. In comparison to other artificial intelligence techniques, deep learning requires a higher amount of data, therefore the number of datasets required for training needs to be known.”

Specific concerns

[1]. Introduction Para1

Terminology is important and there needs to be as clear distinction between the changes in muscle with ageing and muscular disease. Ageing is not a disease. Whilst understanding that some of the changes the occur with ageing also occur in neuro-muscular diseases, the origin of the changes is important needs to be clearly identified.

## Authors’ Response: Thank you for the reviewers’ comment about terminology in ageing. We agree and consider ageing is not a disease. We revise the 1st paragraph int introduction as below.

*** Page: 1, line 36-44

“Muscle diseases is a significant public health issue known to increase mortality and morbidity risk factors [1]. Because of this risk, high rates of muscle disease investigation are common in clinical practice. Bruyere et al. described muscle problems in clinics, 53.30% assessed muscle mass daily in clinical practice, 54.50% muscle strength problems, and around 60% of geriatricians, endocrinologists, and rheumatologists estimate muscle mass [2]. The massive amount of muscle disease investigations in clinical practice can be processed with ultrasound devices' assistance, which has responsiveness and precision of detecting muscles disease in the range of 90 to 95% [3]. Muscle ultrasound is a beneficial contributor to the muscle disease toolkit in clinical and research environments [4].”

[2]. Introduction Para 2

A clear difference between ultrasound images and other medical images, particularly 3d images, is that the transducer is hand-held and operator skill plays a substantial role in the quality of the images obtained. How this affects the complexity and success of Deep Learning routines needs to be part of the introduction.

## Authors’ Response: Thank you for the reviewers' comments on problems with ultrasound, especially 3D images. We realize this can provide background problems in the introduction section. We also encountered challenges in the 1D-4D ultrasound image mode, and we have combined it to strengthen our introduction part.

*** Page: 2, line 47-57

“Additionally, ultrasound characteristics have posed specific challenges, in 1D [6], 2D[7], 3D [8] and 4D [9] images such as image quality due to noise, scarce data, lousy im-age quality, and hand-held and operator skill plays a substantial role which may have an influence on the efficacy of deep learning performance compare with X-Ray, Computed tomography scan (CT-Scan), or Magnetic resonance imaging (MRI) images [10]. Because it requires special attention and resources in overcoming it in the deep learning process. Furthermore, the challenges in muscle ultrasound imaging process were include identification of landmarks, reliability testing, muscle site, image acquisition and analysis, equipment use, normative data and interpretation of results [11,12]. Limitation on muscle ultrasound can be overcome with good training for clinical personnel and clinical experience [4]. Since it is frequently used in clinics, painless and noninvasive [13].”

[3]. Section 2.2

Were the discrepancy and disagreement resolution procedures used and if so, how many times and what was the principal need necessitating their being invoked?

## Authors’ Response: We agree with the reviewer’s comment in discrepancy and disagreement procedure and principals invoked. We have revised the sentences.

*** Page: 4, line 130-136

“Two researchers separately analyzed the results and addressed discrepancies by discussion (P.A. and C.L.). The abstracts and full texts of the related papers have been thoroughly read, and only those that meet the requirements have been chosen. There were ten studies that met the criteria in the results. The researchers then confirmed the selected papers and addressed any disagreements; if any disagreement remained, a third researcher was invited to discuss it, and the findings were evaluated (Y.L.). There was no disagreement in the selection of papers in this review.”

[4]. Figure 4

This figure is misleading - 10% = 1 study as there are only 10 included in the review and should be redrawn with the number of studies shown.

## Authors’ Response: We agree with the reviewer’s comment about misleading figure. We have revised the figure as attached.

[5]. Figure 5

The text is too small to read.

## Authors’ Response: Thank you for reviewer’s comment. The text has increased.

[6]. Section 3.3

An explanation and example of Dice outputs would be helpful.

Authors’ Response:

Thanks for reviewer for mention about Dice explanation in section 3.3, we consider it would be helpful to our study. We agree segmentation is often requires dice. Dice is considered better because it does not only map the actual pixels but accurately marks the boundaries.

*** Page: 6, line 186-188.

“Dice similarity coefficient is common to be used in segmentation task [19]. Dice is also considered superior because it does not only measure the proper marking of the number of pixels but also the precision of boundaries segmentation [44].”

Reviewer 2 Report

In the ultrasound images, skeleton and muscle images are hard to be obtained due to ultrasound characteristics. Authors showed the study of the trends and challenges of deep learning performances about those images. Those topics review paper could be interesting paper because I do not see such review paper. Therefore, it is worthwhile to be published. Authors well summarized the ultrasound images with respect to the skeletal and muscle ultrasound images only for deep learning based techniques. 
However, authors need to add some missing references and previous paper reviews. Figure qualities should be improved. Therefore, the manuscript could be minor revision if authors follows the comments.

1. In the reference section, authors should use abbreviated journal names.
2. Please provide funding information. Please do not provide that in acknowledgments.
3. Please increase Figure 1 label sizes.
4. Please improve Figure 3 labels.
5. Please increase Figure 5 label sizes.
6. Please add the reference (A deconvolutional layer without a concatenation layer can be used for segmentation in skeletal muscles) with the reference (Hashimoto, Fumio, et al. "Automated segmentation of 2D low-dose CT images of the psoas-major muscle using deep convolutional neural networks." Radiological physics and technology 12.2 (2019): 210-215. ) or another reference.
7.Please add the reference (A concatenation layer is required to improve precision sharpness, which is followed by a convolution layer to acquire the final output to overcome the problem ) with the reference (Zhang, S., Zhang, S., Zhang, C., Wang, X., & Shi, Y. (2019). Cucumber leaf disease identification with global pooling dilated convolutional neural network. Computers and Electronics in Agriculture, 162, 422-430. ) or another reference.
8.Please add the reference (CNN and Deep CNN have popularly used base architecture for classification in muscle ultrasound images ) with the reference (Liu, Shengfeng, et al. "Deep learning in medical ultrasound analysis: a review." Engineering 5.2 (2019): 261-275. ) or another reference.
9.Please add the reference (Deep learning is an upcoming solution for the detection of landmarks ) with the reference ( Riaz, H.; Park, J.; Choi, H.; Kim, H.; Kim, J. Deep and Densely Connected Networks for Classification of Diabetic Retinopathy. Diagnostics 2020, 10, 24).
10.Please add the reference (Skeletal muscle ultrasound is the largest area of deep learning research in muscle ultrasound images. ) with the reference ((Liu, Shengfeng, et al. "Deep learning in medical ultrasound analysis: a review." Engineering 5.2 (2019): 261-275.  ) or another reference.
11. Please use formal expression (Also -> In addition).
12. Figure 4 label sizes need to be reduced.

Author Response

Comments and Suggestions for Authors

In the ultrasound images, skeleton and muscle images are hard to be obtained due to ultrasound characteristics. Authors showed the study of the trends and challenges of deep learning performances about those images. Those topics review paper could be interesting paper because I do not see such review paper. Therefore, it is worthwhile to be published. Authors well summarized the ultrasound images with respect to the skeletal and muscle ultrasound images only for deep learning based techniques. 
However, authors need to add some missing references and previous paper reviews. Figure qualities should be improved. Therefore, the manuscript could be minor revision if authors follows the comments.

## Authors’ Response: Thank you very much for your thoughtful review of our manuscript. As reviewer mention, In the ultrasound images, skeleton and muscle images are hard to be obtained due to ultrasound characteristics. Including image quality due to noise, scarce data, lousy images, and hand-held and operator skill plays a substantial role.

[1]. Abbreviated journal names

[1]. In the reference section, authors should use abbreviated journal names.

## Authors’ Response: Thanks for your reminding, we added abbreviated journal name in the reference section.

[2]. Funding information

  1. Please provide funding information. Please do not provide that in acknowledgments.

## Authors’ Response: Thanks for your reminding, we have move funding information from acknowledgements to funding section.

[3]. Figure 1

  1. Please increase Figure 1 label sizes.

## Authors’ Response: Thank you for reviewer suggestion, the label has increased as shown attached.

[4]. Figure 3

  1. Please improve Figure 3 labels.

## Authors’ Response: Figure 3 labels has improved.

[5]. Figure 5.

  1. Please increase Figure 5 label sizes.

## Authors’ Response: Thank you for reviewer suggestion, the label has increased.

[6]. Please add the reference

  1. Please add the reference (A deconvolutional layer without a concatenation layer can be used for segmentation in skeletal muscles) with the reference (Hashimoto, Fumio, et al. "Automated segmentation of 2D low-dose CT images of the psoas-major muscle using deep convolutional neural networks." Radiological physics and technology 12.2 (2019): 210-215. ) or another reference.
    ## Authors’ Response: Thank you very much for reviewer suggestion, the reference had added in line 221-222 to strengthen the statement.

*** Page: 8, line 221-222.

“A deconvolutional layer without a concatenation layer can be used for segmentation in skeletal muscles [50].”

[7]. Please add the reference

  1. Please add the reference (A concatenation layer is required to improve precision sharpness, which is followed by a convolution layer to acquire the final output to overcome the problem ) with the reference (Zhang, S., Zhang, S., Zhang, C., Wang, X., & Shi, Y. (2019). Cucumber leaf disease identification with global pooling dilated convolutional neural network. Computers and Electronics in Agriculture, 162, 422-430. ) or another reference.
    ## Authors’ Response: Thank you very much for reviewer suggestion, Zhang et. al (2019) “Cucumber leaf disease identification with global pooling dilated convolutional neural network.” have a beneficial information to strengthening our statement in concatenation layer part. The reference had added in line 249-251.

*** Page: 8, line 249-251.

. A concatenation layer is required to improve precision sharpness, which is followed by a convolution layer to acquire the final output to overcome the problem [53].”

[8]. Please add the reference

  1. Please add the reference (CNN and Deep CNN have popularly used base architecture for classification in muscle ultrasound images ) with the reference (Liu, Shengfeng, et al. "Deep learning in medical ultrasound analysis: a review." Engineering 5.2 (2019): 261-275. ) or another reference.
    ## Authors’ Response: Thank you very much for reviewer suggestion, the paper of Liu et.al (2019) has given us more insight and strengthening our statement in network architecture on medical ultrasound images. The reference had added in line 306-307. CNN and Deep CNN have popularly used base architecture for classification in muscle ultrasound images [43].”

[9]. Please add the reference

  1. Please add the reference (Deep learning is an upcoming solution for the detection of landmarks ) with the reference ( Riaz, H.; Park, J.; Choi, H.; Kim, H.; Kim, J. Deep and Densely Connected Networks for Classification of Diabetic Retinopathy. Diagnostics 2020, 10, 24).
    ## Authors’ Response: Thank you very much for reviewer suggestion, the study of Riaz et.al 2020 has given us more input about deep learning task in landmarks detection. The reference had added in line 58-59. “ Deep learning is an upcoming solution for the detection of landmarks [14]”

[10]. Please add the reference

  1. Please add the reference (Skeletal muscle ultrasound is the largest area of deep learning research in muscle ultrasound images. ) with the reference ((Liu, Shengfeng, et al. "Deep learning in medical ultrasound analysis: a review." Engineering 5.2 (2019): 261-275.  ) or another reference.

## Authors’ Response: Thank you very much for reviewer suggestion, the paper of Liu et.al (2019) has beneficial to us and strengthening our statement in skeletal muscle is largest area in deep learning task. The reference had added in line 179-180. “Skeletal muscle ultrasound is the largest area of deep learning research in muscle ultrasound images [43]..”

[11]. Formal expression

  1. Please use formal expression (Also -> In addition).
    ## Authors’ Response: Thank you reviewer for the suggestion, the formal expression had been used.

[12]. Figure 4

  1. Figure 4 label sizes need to be reduced.

## Authors’ Response: Thank you reviewer for the suggestion, The Figure 4 label sizes have been reduced.

Reviewer 3 Report

In this review paper the authors aim to study the challenges and trends of deep learning performance, especially to overcome muscle ultrasound image problems such as low-quality images, muscle movement in skeletal and muscle thickness in smooth muscles. The authors reviewed about skeletal muscle segmentation and classification using deep learning algorithms. The authors highlight the following challenges and findings: In skeletal muscle segmentation are muscles that move regularly, and modified layers in upsampling are needed to keep data information recorded. In skeletal muscle classification, the problems faced are area-specific, thus making a cropping strategy useful. There is no need to add additional layer modifications in smooth muscle segmentation because muscle thickness is the major problem.

The manuscript is interesting. However, the following major concerns need to be addressed.

  1. Abstract need to be improved. Specifically rewrite the sentences in Line 17-18(first two lines) and Line 26-28 “The challenges…….recorded”. Further, include about findings.
  2. Page 4, subsection 2.3. Data analysis: Give more details about data analysis with respect to Table1. What are the evaluation metrics used in output and what is image solution?
  3. Page 9, Subsection 3.9. Dataset requirements: How the datasets are obtained from different studies? Give more details about datasets.
  4. In limitations section (3.11), discuss in detail about future work to overcome those existing limitations.
  5. Improve conclusion section. Discuss the advantages and possible real time applications of utilizing deep learning for Skeletal and Smooth Muscle Ultrasound Images.
  6. In introduction, Line 56-60 include other applications like computerized liver cancer diagnosis and provide references. Refer the following paper for details, "Computer aided diagnostic system for ultrasound liver images: a systematic review”.

Author Response

Comments and Suggestions for Authors

In this review paper the authors aim to study the challenges and trends of deep learning performance, especially to overcome muscle ultrasound image problems such as low-quality images, muscle movement in skeletal and muscle thickness in smooth muscles. The authors reviewed about skeletal muscle segmentation and classification using deep learning algorithms. The authors highlight the following challenges and findings: In skeletal muscle segmentation are muscles that move regularly, and modified layers in upsampling are needed to keep data information recorded. In skeletal muscle classification, the problems faced are area-specific, thus making a cropping strategy useful. There is no need to add additional layer modifications in smooth muscle segmentation because muscle thickness is the major problem.

The manuscript is interesting. However, the following major concerns need to be addressed.

## Authors’ Response: Thank you very much for your thoughtful review of our manuscript. As the reviewer mentioned in the general comment, we agree and realize that some improvements are needed to strengthen our study.

[1]. Abstract

  1. Abstract need to be improved. Specifically rewrite the sentences in Line 17-18(first two lines) and Line 26-28 “The challenges…….recorded”. Further, include about findings.

## Authors’ Response: Thank you very much for reviewer suggestion, we have rewritten the sentences in line 17-18 and 26-28.

“Deep learning has aided in the improvement of diagnosis identification, evaluation, and interpretation of muscle ultrasound images, which may benefit clinical personnel.” And “Skeletal muscle segmentation presents difficulties due to the regular movement of muscles and noise, to keep data information recorded, skip connections and modified layers in upsampling are needed.”

[2]. Evaluation metrics

  1. Page 4, subsection 2.3. Data analysis: Give more details about data analysis with respect to Table1. What are the evaluation metrics used in output and what is image solution?

## Authors’ Response: We agree with the reviewer’s comment about data analysis respect to Table1. We consider explaining more detail about evaluation metrics output until image solution is beneficial to our study. We have revised the sentences as bellow.

*** Page: 4, line 139-147

“The findings were divided into three groups: skeletal muscle segmentation, smooth muscle segmentation, and skeletal muscle classification (Figure 1). Besides, the following data are summarized: 1) Muscle type, which provides information about the skeletal or smooth muscle type. 2) Deep learning approach for the segmentation or classification of ultrasound images. 3) Citations, containing author information and the year the article was published. 4) Research objectives that cover specific muscle areas. 5) Deep learning architecture method 6) Dataset as a reference for deep learning model studying ultrasound images. 7) Output from matrix performance to see deep learning results. 8) Image solutions to determine the method used to solve problems in muscle ultrasound images to improve deep learning performance from 2018 to 2020 (Table 1).”

[3]. Dataset requirements

  1. Page 9, Subsection 3.9. Dataset requirements: How the datasets are obtained from different studies? Give more details about datasets.

## Authors’ Response: Thank you for reviewer comment about dataset explanation in subsection 3.9. we consider the obtained of dataset will be strengthening our study. The 10 studies using the same method with Ultrasound B-Mode images. We added the explanation as bellow.

*** Page: 10, line 335-344

“The dataset used of the ten studies was derived from capturing Ultrasound B-Mode images. In addition, the use of segmentation on ultrasound B-Mode in thyroid nodules can be used in real-time [66]. It allows the possible application of real-time deep learning to muscle ultrasound images.

Difficulties with the number of images were common challenges among the two studies which did not use the augmentation or transfer learning technique in muscle ultrasound images. In comparison to other artificial intelligence techniques, deep learning requires a higher amount of data, therefore the number of datasets required for training needs to be known.”

[4]. limitations section

  1. In limitations section (3.11), discuss in detail about future work to overcome those existing limitations.

## Authors’ Response: Thank you for reviewer comment about our future work, we consider that added the future work are important, we had revised the limitation as bellow.

*** Page: 11, line 359-369

“3.11. Limitation

There are two limitations in this study. The first limitation is that the muscle segmentation and classification research use data augmentation or transfer learning because of muscle ultrasound image problems. However, both solutions currently still have disadvantages and need further investigation [71]. Future work will discover additional solutions for muscle ultrasound image problems such 4D ultrasound, on-fly-data augmentation for 3D images, and active learning as an alternative for augmentation or transfer learning to overcome the limitation [72,73]. The second limitation is that this study is limited by two types of muscle: skeletal muscle and smooth muscle, without examining the cardiac muscle. The difficulty with cardiac muscle ultrasound images is sequential muscle data. A deep learning model which can read sequential data is needed, such as Long short-term memory (LSTM) [74,75].”

[5]. Conclusion

  1. Improve conclusion section. Discuss the advantages and possible real time applications of utilizing deep learning for Skeletal and Smooth Muscle Ultrasound Images.

## Authors’ Response: Thank you for reviewer suggestion to mention possibility real time application in the conclusion, we agree with it and the conclusion section had improved.

*** Page 11, line 371-380

“Deep learning is an effective strategy for determining skeletal and smooth muscle conditions to help clinic personnel in landmark identification, muscle site, and reliability testing using segmentation or classification via ultrasound images. The various challenges in skeletal muscle segmentation are muscles that move regularly and noise, thus skip connections and modified layers in up sampling are needed to keep data information recorded. There is no need to add additional layer modifications in smooth muscle segmentation because muscle thickness is the main problem. In the skeletal muscle classification, the problems faced are area-specific, thus making a cropping strategy useful. The use of segmentation on ultrasound B-Mode allows the possible application of real-time deep learning to muscle ultrasound images.”

[6]. Introduction

  1. In introduction, Line 56-60 include other applications like computerized liver cancer diagnosis and provide references. Refer the following paper for details, "Computer aided diagnostic system for ultrasound liver images: a systematic review”.

## Authors’ Response: We agree with the reviewer’s comment and have revised and added the references to strengthening our statement in introduction section.

Jabarulla et al. (2017). “Computer aided diagnostic system for ultrasound liver images: a systematic review” is benefit for complement information about liver identification and diagnoses using deep learning in our introduction part.

*** Page: 2, line 68-72

“Deep learning in ultrasound images has been used to support clinical tasks, starting in 2015 [20], such as identification of the abdominal plane [21,22], frame labeling on fetal pregnancy images [23], analyzing abdominal circumference [24], vessel detection [25], tongue contour extraction [26], carotid plaque classification [27], thyroid nodule detection [28] and identification liver cancer [29].”

Round 2

Reviewer 1 Report

None

Reviewer 3 Report

After reading through, I felt the manuscript well revised and addressed all the responses appropriately.